# Detection of a Novel Chlamydia Species in Invasive Turtles

**DOI:** 10.3390/ani12060784

**Published:** 2022-03-20

**Authors:** Laura Bellinati, Stefano Pesaro, Federica Marcer, Patrizia Danesi, Alda Natale, Letizia Ceglie

**Affiliations:** 1Istituto Zooprofilattico Sperimentale delle Venezie, 35020 Legnaro, Italy; pdanesi@izsvenezie.it (P.D.); anatale@izsvenezie.it (A.N.); lceglie@izsvenezie.it (L.C.); 2Dipartimento di Scienze Agro-Alimentari, Ambientali e Animali Sezione di Patologia Veterinaria, Università degli Studi di Udine, Via Sondrio, 33100 Udine, Italy; stefano.pesaro@uniud.it; 3Dipartimento di Medicina Animale, Produzioni e Salute, Università degli Studi di Padova, 35020 Legnaro, Italy; federica.marcer@unipd.it

**Keywords:** *Chlamydia*, genome analysis, potential zoonosis

## Abstract

**Simple Summary:**

The pond slider (*Trachemys scripta*) is a turtle species native to Central America. Pond sliders have been commercialized as pets since the 1950s, but often ended up being released or escaping into an environment to which they are allochthonous. *Trachemys scripta* is presently classified as an invasive alien species in Europe and other countries. The introduction of pond sliders in foreign ecosystems has had a deep environmental and ecological impact. Moreover, freed or escaped captive turtles could be carriers of pathogens, such as *Chlamydiaceae*. In this study, we report the identification of a *Chlamydia* spp. in two pond sliders found dead after the hibernation period. Further molecular characterization revealed that this *Chlamydia* is identical to a strain reported in Poland in 2017, found in both captive and free-living freshwater turtles, and has a close phylogenetic relationship to *Chlamydia pneumoniae*, a species infectious to humans, and to chlamydial strains found in other reptiles. This first finding evidences the presence of this novel *Chlamydia* in Italian turtles, but further studies will be necessary to evaluate the prevalence in the local turtles’ population and the strain pathogenicity.

**Abstract:**

*Trachemys scripta* is a turtle species native to Central America. Since the 1950s, pond sliders have been imported worldwide as companion animals, but have often ended up in foreign ecosystems with great ecological consequences. Moreover, both autochthonous and invasive species of turtles can be carriers of pathogens, including *Chlamydiaceae*. In the present study, pulmonary tissues collected from four *Trachemys scripta* were tested with a 23S-targeting real-time PCR (rPCR) specific for the *Chlamydiaceae* family. The turtles were hosted in a rescue center for wild exotic animals located in northeastern Italy, and were found dead after the hibernation period. Two out of four individuals resulted positive in rPCR for the presence of *Chlamydiaceae*. Further characterization of this positivity was performed by phylogenetic analysis of the 16S rRNA and outer membrane protein A genes. The phylogenetic tree showed that these chlamydial strains are identical to a novel *Chlamydia* reported in 2017 in Polish freshwater turtles, and closely related to *Chlamydia pneumoniae* and to other chlamydial strains found in reptiles. This first finding evidences the presence of this *Chlamydia* strain in Italian turtles, but further studies will be necessary to confirm the presence and the strain pathogenicity and to evaluate its prevalence in the local turtles’ population.

## 1. Introduction

The pond slider (*Trachemys (T.) scripta*) is a turtle species native to the southern United States and northern Mexico. Since the 1950s, pond sliders have become very popular as pets, and a high number of specimens have been captured and exported to Europe and other countries as companion animals [1]. However, when captive turtles grow out of their enclosure or their caretakers become tired of them, they are often released or escape into the wild. Hence, there have been reports of the presence of the *T. scripta* specimen in many European countries, but also in Africa, South America and Asia [2,3,4,5], where this species became an opportunistic inhabitant of freshwater habitats such as rivers, lakes and natural or public ponds, generally close to the human dwellings where they were released. Moreover, reproduction of pond sliders in the natural environment has been reported in many European countries with Mediterranean climate conditions [6], including Italy [7,8].

The threats arising to the ecosystem from the introduction of invasive alien species (IAS) are well documented [9,10]. Pond sliders compete directly with native turtles species for food and basking places in wetlands, where they coexist [7,11]; moreover, they prey on smaller animals and plants, which influences the ecosystem and the aquatic communities [12,13] and causes a great ecological impact on the environment. For this reason, the European Union included *T. scripta* in the list of Invasive Alien Species of Union concern [14], and since 2016 it has been forbidden to import, breed, commercialize or release this species across the whole of the European Union [15]. Despite the ban, the specimen already released in the natural environment resulted in a serious threat to native turtle species and the biodiversity, especially in the European countries with a Mediterranean climate, where the natural reproduction of *T. scripta* has been reported.

Another concern about the introduction of IAS is their possible role as carriers of pathogens [16,17], as their invasion into foreign environments may contribute to the spread of infectious diseases, becoming a possible threat to both animal and human health. Like other turtles, *T. scripta* have been found to carry and contribute to the spread of different pathogens, including parasites such as nematodes and trematodes [18,19], pathogenic fungal species [20], herpesvirus [21] and bacteria such as *Mycoplasma* [21,22] *Enterobacteriaceae*, including *Salmonella* [23,24,25] and *Chlamydiaceae* [26,27,28].

*Chlamydiaceae* is a family of Gram-negative intracellular obligate bacteria able to infect a wide range of hosts [29] and are the causative agents of chlamydiosis in humans, domestic animals and wildlife. The *Chlamydiaceae* family belongs to the order *Chlamydiales* and is characterized by an intrinsic high genetic diversity, with a taxonomic classification constantly evolving. Until 2021, *Chlamydiaceae* consisted of the single genus *Chlamydia* (*C.*), but recent findings in flamingoes have enriched the family of the new genus *Chlamydiifrater* gen. nov. with two new species, *Chlamydiifrater phoenicopteri* sp. nov. and *Chlamydiifrater volucris* sp. nov. [30]. The genus *Chlamydia* consists of 14 characterized species, five of which have been fully characterized through whole-genome sequencing in the last decade, specifically *C. avium*, *C. gallinacea*, and *C. buteonis,* identified in birds, and *C. serpentis* and *C. poikilothermis* identified in snakes. Moreover, four new candidate species have been recently proposed after molecular detection and identification [31].

Since their first discovery in 1944 in a lizard, *Chlamydiaceae* have been observed among many different reptiles, discovered thanks to the use of new molecular methods [31]. The presence of *C. pneumoniae*, infectious to humans as well [26,27,32,33,34,35,36], was confirmed in different types of reptiles, together with strains similar to *C. pneumoniae* and *C. caviae.* This led to the classification of three new taxa related to *C. pneumoniae**, C. serpentis* and *Candidati*, C. corallus and C. sanzinia, and one similar to *C. caviae, C. poikilothermics* [32,37,38]. Chlamydial infections show a variety of clinical manifestations, ranging from inapparent to clinical disease mostly in the urogenital or the respiratory tracts and conjunctivitis. In reptiles, infection of *Chlamydia* spp. has been found with and without clinical signs [26,27,28,33,34,35,36], but the pathogenicity of new proposed species is still under investigation. Moreover, its zoonotic potential cannot be ruled out, considering the phylogenetic relationship with *C. pneumoniae* [39,40]. In turtles, a new species phylogenetically similar to *C. pecorum* was identified in Spain, Poland and Germany [26,27,28] and designated as *Candidatus* C. testudinis [28], due to the lack of an isolate. This chlamydial strain seems to contribute or cause ocular disease and nasal discharge. Conversely, other *Chlamydia* have been found in turtles without clinical signs [27], but there is still limited bibliography available on the presence and possible risks associated with *Chlamydiaceae* in turtles.

With the aim of identifying the cause of mortality in four *T. scripta* found dead after the hibernation period in northeastern Italy, we performed necropsy and screening assays for the presence of pathogens, including *Chlamydia* spp. PCR-positive samples were submitted to molecular characterization.

## 2. Materials and Methods

### 2.1. Sampling

Four mature *Trachemys scripta,* kept in a confined area in the wildlife and exotic animal rescue center located in Grado lagoon (northeastern Italy), were found dead in April 2019, after the winter hibernation period. The turtles were submitted to necropsy and to routine diagnostic investigations for the presence of common fungi and parasites. Lung tissues samples were collected for molecular investigation on the presence of bacteria belonging to from the *Chlamydiaceae* family.

### 2.2. Molecular Analysis

#### 2.2.1. Nucleic Acid Extraction

DNA isolation from the collected samples was performed as validated in our laboratory for routine diagnostic. Briefly, lung tissues collected from the pond sliders were homogenized at a 1:10 dilution in 600 µL of PBS, with TissueLyser II (QIAGEN, Hilden, Germany). DNA extraction was performed from 200 µL of lung tissue homogenate with High Pure PCR Template Preparation Kit (Roche Diagnostics, Mannheim, Germany) according to the manufacturer’s instructions. To check the efficiency of the DNA extraction, monitor the effect of any potential inhibition and validate each negative result, a universal heterologous control, DNA Intype IC-DNA (Indical Bioscience GmbH, Leipzig, Germany), was added to each sample in the extraction step with a ratio of 1:10 of the total elution volume.

#### 2.2.2. Real-Time PCR

To detect the presence of bacterial DNA of the *Chlamydiaceae* family, a screening real-time PCR targeting a 111 bp fragment of the 23S rRNA gene was performed according to Ehricht et al. [41]. Real-time PCR was conducted in a total volume of 15 µL containing 2 µL of extracted DNA and 7.5 µL of 2X Path-ID qPCR Master Mix (Thermo Fisher Scientific, Waltham, MA, USA). A duplex protocol was applied with 625 nM of each primer and 100 nM of a 5′FAM-3′TMR probe targeting 23S rRNA *Chlamydiaceae* gene, and 100 nM of each primer and probe specific for the exogenous internal control, as described in the OIE Manual for Avian Chlamydiosis [42]. The temperature–time profile was 95 °C for 10 min, 45 cycles of 95 °C for 15 s and 60 °C for 60 s.

DNA isolated from *C. abortus* culture and DNase-RNase free water were used as positive and negative controls for the reaction mix in each run, respectively. Samples with Ct values superior to 40.0 were considered negative.

#### 2.2.3. Species Designation

In order to identify the chlamydial species, the samples that had tested positive to the real-time PCR were characterized by sequencing a 16S rRNA gene portion of 278 bp with the primers set described by Vicari et al. [43]. The FastStart™ Taq DNA Polymerase kit (Roche Diagnostics, Mannheim, Germany) was used with 5 µL of extracted DNA as template in a final volume of 25 µL with the following final concentrations: 2.5 mM of MgCl_2_, 0.2 mM of each dNTPs and 0.4 µM of each primer. After the initial denaturation step at 95 °C for 15 min, a touchdown protocol was applied to 20 cycles at 95 °C for 30 s, a decreasing annealing temperature from 65 °C to 55 °C for 30 s and 72 °C for 30 s, followed by 25 cycles with the same conditions and annealing step at 55 °C. A final extension step at 72 °C for 5 min was performed. PCR-amplified segments were purified with ExoSAP-IT Express PCR Product Cleanup (Applied Biosystems, Foster City, CA, USA) and sequenced with Brilliant Dye™ Terminator Cycle Sequencing kit Big Dye^®^ Terminator v3.1 Cycle Sequencing (Nimagen, Nijmegen, The Netherlands) on a the ABI PRISM^®^ 3130xl Genetic Analyzer (Applied Biosystems, Foster City, CA, USA).

Sequencing data were subjected to BLAST search against the NCBI database to identify the most related sequences.

### 2.3. Phylogenetic Analysis

In order to perform an additional phylogenetic characterization of the positive samples, a 920 bp fragment of the *outer membrane protein A* (*ompA*) and a 1200 bp fragment of the 16S rRNA were sequenced. The fragments were amplified using primer pairs CTU/CTL for *ompA* gene [44], and 16R/S1 for the 16S rRNA [45]. The FastStart™ Taq DNA Polymerase kit (Roche Diagnostics, Mannheim, Germany) was used with 5 µL of extracted DNA in a final volume of 25 µL with the following final concentrations: 2.5 mM of MgCl_2_, 0.2 mM of each dNTPs and 0.4 µM of each primer, and the addition of 5 µL GCRICH Buffer. Temperature–time for both primer sets was 94 °C per 5 min, 5 cycles at 94 °C for 45 s, 48 °C for 45 s and 72 °C for 90 s, followed by 40 cycles at 94 °C for 45 s, 50 °C for 45 s and 72 °C for 90 s, and a final annealing step at 72 °C per 7 min. PCR-amplified segments were purified and sequenced as described in Section 2.2.3.

Sequencing data were subjected to BLAST search against the NCBI database to identify the most related sequences. To evaluate the phylogenetic relationship between *Chlamydia* spp. and the tested samples, the nucleotide sequences were aligned by mafft v7 software [46] with public sequences of importance for phylogenetic and epidemiologic consideration. The MrBayes method was applied for the construction of the phylogenetic tree, using MrBeast software [47]. Based on MEGA 10 calculations, the general time-reversible (GTR) model was chosen for the analysis of 16S rRNA gene, whereas, for the *ompA* gene, the Hasegawa–Kishino–Yano (HKY85) model was chosen. Both DNA substitution models were applied with gamma-distributed rate variation across sites. Posterior probabilities were calculated as a measure of branch support.

## 3. Results

### 3.1. Sampling

Post-mortem examination of the four pond sliders did not show any significant findings. Other tests performed on the four specimens reported no evidence for the presence of parasite and mycological infection. Histological data on lung tissues are not available because specimens were autolytic and their conservation status was not ideal for this analysis.

### 3.2. Molecular Analysis

DNA extracted from lung homogenates of two turtles tested positive in the *Chlamydiaceae*–specific real-time PCR, with Cq values of 38.4 and 35.4.

Amplification of the 278 bp 16S rRNA gene portion for the species designation was successful for both positive samples. The two sequences obtained turned out to be identical. The BLAST analysis carried out in the GenBank database revealed a 100% nucleotide homology with four published sequences of uncultured *Chlamydia* spp. found in free-living and captive freshwater turtles in Poland by Mitura et al. [27] in 2016.

### 3.3. Phylogenetic Analysis

The complete amplification of the *ompA* and 16S rRNA genes was performed on both positive samples. However, full sequences were achieved only for the sample with a Cq value of 35.4, while for the other sample the Cq value was likely to be too low to obtain full sequences. BLAST analysis confirmed the close relationship with the *Chlamydiaceae* found by Mitura et al. [27], with a 100% nucleotide homology for the *ompA* gene and a 99.7% homology for the 16S rRNA gene. The sequences were submitted to GenBank and are available with accession number OM055817 for the 16S rRNA gene, and OM055818 for the *ompA* gene.

The phylogenetic tree constructed on the basis of the available 16S rRNA sequences is presented in Figure 1a and shows a topology almost identical to the phylogenetic tree built on the *ompA* gene sequences and presented in Figure 1b. The sequence of the chlamydial strain identified in this study forms a monophyletic clade with ten sequences of *Chlamydia* spp. found in free-living *T. scripta*, captive *T. scripta* and *Emys orbicularis* hatchlings identified by Mitura et al. [27] in Poland in 2016. The clade is supported by a posterior probability value of 100. To our knowledge, no other sequences of *Chlamydia* spp. belonging to this clade have been identified.

This new *Chlamydia* taxon is most closely related to *C. pneumoniae*, a strain with a known zoonotic potential, with a percentage of identity for the *ompA* gene sequence of 75.4%, and to other *Chlamydia* spp. found in reptiles and with unknown pathogenicity, *C. serpentis (73*.43%) and *Candidatus* C. sanzinia (71.9%) and C. corallus (73.7%). Moreover, there is high similarity (between 72% and 74%) with the clade composed by *C. pecorum* and sequences identified in Mitura et al. [27] that have been classified in the proposed *Candidatus* C. testudinis [28].

In the phylogenetic tree based on the *ompA* gene (Figure 1b), more specifically in the clade that includes the *Chlamydia* spp. identified in our study, three sub-branches can be identified, and this division is supported by posterior probability. The subgroup that includes the sequence obtained in this work consists of four other sequences found by Mitura et al. [27] in three free-living pond sliders and one captive pond slider. Only in one free-living specimen that was found in poor health conditions and died during the quarantine period was the chlamydial positivity detected in tissue samples, while all the other instances of positivity were found only in the swab specimen.

## 4. Discussion

The threats arising from IAS turtles are well documented, but there are still few data available on *Chlamydiaceae* occurrence in these species, and most published research studies concerning reptiles include only animals living in captivity in farms, zoos or private collections [26,36,48,49]. In addition, the source of chlamydial infections in reptiles often remains unknown [29].

In this study, we found *Chlamydia*-positive lung tissues in two *T. scripta* individuals, known as IAS turtles, found dead after the winter hibernation period. However, the low Cq values observed could be due to the poor conditions of the collected specimen or suggest a low chlamydial shedding. In light of this, whether the chlamydial infection was the main cause of the animals’ death is still unknown. The available literature suggests that it is more likely that other factors triggered the systemic infection and death. These include temperature changes, stress, malnutrition caused by hibernation, or other pathogens [50,51] combined with the *Chlamydia* colonization. This hypothesis is supported by Mitura et al. [27], where the same chlamydial strain detected in this work was detected in a tissue sample collected from a free-living pond slider found in poor overall condition and deceased during the quarantine period. Conversely, only swab samples resulted positive for *Chlamydia* spp. in healthy *T. scripta*.

The *Chlamydia* spp. characterized in this study is identical to a chlamydial strain found for the first time by Mitura et al. [27] in a survey conducted in 2016 to evaluate the prevalence of *Chlamydiaceae* in turtles in Poland. Besides, the chlamydial strain detected in our study was found in both free-living and captive pond sliders, as well as in a breeding facility where a high mortality in hatchlings of the *Emys orbicularis* species occurred. In the facility, this *Chlamydia* was found in *Emys orbicularis* hatchlings, and the same chlamydial strain was detected in *T*. *Scripta* turtles showing no clinical signs. The animals were kept in separate tanks but were handled by the same personnel, most likely the vector causing the pathogen to spread in the facility. Conversely, this strain was not found in the *Emys orbicularis* adult living in the same facility, and neither in free-living nor captive *Emys orbicularis* originating from other locations.

The finding in our work referring to the same *Chlamydia* strain suggests that this new taxon is present in turtles both in Poland and in Italy. This new species forms a separate clade closely related to *C. pneumoniae* and other *Chlamydia* identified in reptiles, such as *C. serpentis* and *Candidatus* C. sanzinia, C. corallus and C. testudinis. Extended investigation will be necessary to confirm the presence and evaluate the prevalence of this novel strain in the native turtle’s population in Italy. It will be also important to acquire information about whether autochthonous turtles’ species are harbouring *Chlamydia* spp. or if IAS turtles carry these bacteria and can further threaten protected species.

Moreover, the zoonotic implications of this newly discovered species cannot be excluded and should be investigated; so far, our knowledge of the *Chlamydia* described is insufficient to determine its zoonotic potential. Many previous studies have drawn attention to the zoonotic risks associated with reptiles kept as companion animals [22,24,52], since they may represent a high risk, especially when they do not show any clinical signs. Furthermore, these animals might be released into the wild or escape, joining the IAS population, and posing a threat to public health. Further surveys will be important and necessary to expand our knowledge on this issue.

## 5. Conclusions

The *Chlamydiaceae* family is of great significance for animal and human health, although our knowledge of many of its related aspects is still limited.

In this study, we describe the finding of a novel *Chlamydia* spp. reported for the first time in both captive and free-living freshwater turtles in 2016 in Poland by Mitura et al. [27]. To our knowledge, this study is the first report of its kind in free-living *T. scripta* in Italy.

The limited number of turtles analyzed in this study does not permit one to evaluate the importance of this taxon and other *Chlamydiaceae* in the Italian turtle population. Further studies with a larger sample size, including both wild and captive turtles, will be essential to confirm the presence and evaluate the prevalence of *Chlamydiaceae*. Additionally, data on the Italian turtle and reptile population are limited, but a recent study has identified the presence of *C. pneumoniae* and other *Chlamydiales* in Italian loggerhead sea turtles [53]. The identification of both *C. pneumonia* in sea turtles and a similar taxon described above should be taken into account for public health, since *C. pneumonia* represents a serious threat to human health, being implicated in acute and chronic infections.

The role of turtles as carriers of *Chlamydiaceae* and the possible impacts on animal health and conservation are two issues that still need to be explained, as well as the potential threat to the professional categories working with these reptiles (e.g., wildlife caretakers, veterinarians and herpetologists). Therefore, it is crucial to deeply characterize any Chlamydia hosted by turtles.

## Figures and Tables

**Figure 1 animals-12-00784-f001:**
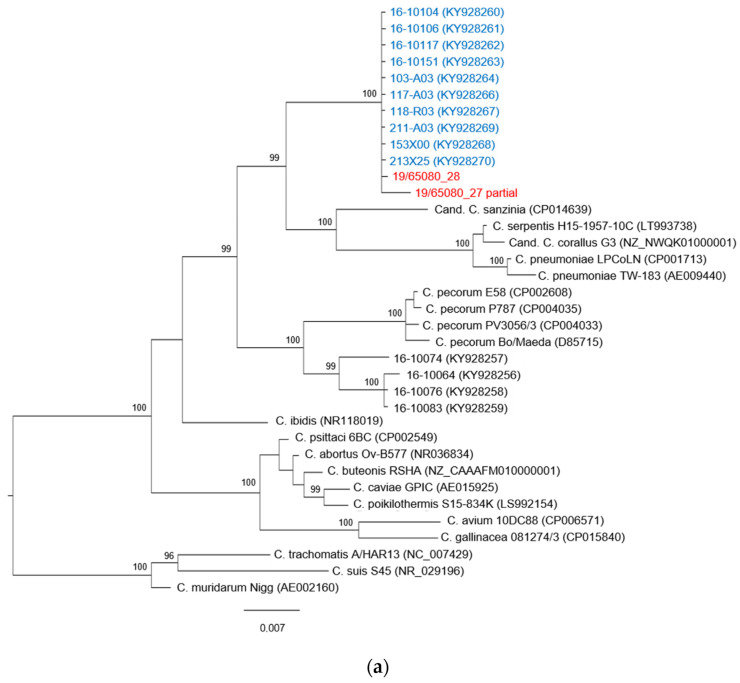
Phylogenetic trees based on (**a**) 16S rRNA gene fragment (920 bp) and (**b**) ompA gene (1200 bp). Representative sequences of *Chlamydiaceae* species as well as strains found in reptilian hosts were included and colored in black. The sequences identified in this study are shown in red, and sequences from Mitura et al. [27] are highlighted in blue. *Chlamydia muridarum* strain Nigg was used as outgroup. The phylogenetic tree was constructed with the MrBayes method, the GTR substitution model was applied for 16S rRNA gene, and the HKY85 model was applied to the *ompA* gene. Posterior probability values are shown as percentages, values under 95 were considered not reliable and are not shown. The scale bar indicates the number of substitutions per site.

## Data Availability

The datasets supporting the conclusions of this article are included within the article. The bacterial nucleotide sequences obtained in this study are openly available in the GenBank database (https://www.ncbi.nlm.nih.gov/genbank/, accessed on 17 June 2021; ID: OM055817 and OM055818).

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
