# Peer review of "Detection of a Novel Chlamydia Species in Invasive Turtles"

_animals, 2022, doi:10.3390/ani12060784_

Round 1
Reviewer 1 Report
To determine the cause of death of four tortoises the authors have examined lung tissue and detected DNA of Chlamydiaceae in two samples. A 278-bp segment of the 16S rRNA gene was sequenced to identify the chlamydial species. Further, a 920-bp fragment of ompA and the nearly complete rDNA were sequenced.
Although the amount of new data presented is modest (3 DNA sequences) the observations and conclusions are important and deserve publication. However, the manuscript contains a number of ambiguities and omissions that should be rectified.
- As a major objection, the term "species" is overused in the manuscript. (Prior to classification as a species, the term "taxon" may also be appropriate.) The authors use the term without having verified at least some of the criteria for species definition set out by the Subcommittee for the taxonomy of chlamydiae. For instance, sequence similarity values of the rDNA sequences to some of the related Chlamydia spp. should be presented. There may also be observations on the tissue samples examined by the pathologist that could have revealed typical features of a chlamydial colonization.
- Line 65: The description of the current status of the family Chlamydiaceae is incomplete. The genus Chlamydia currently comprises 18 species, among them four at Candidatus rank. In addition, the genus Chlamydiifrater (Cf.) with the two species Cf. phoenicopteri and Cf. volucris was recently introduced (see paper by Vorimore et al. Syst Appl Microbiol. 2021Jul;44(4):126200). Please, update the text.
- Line 80: The word "seldom" should be deleted.
- Line 81: Candidatus in Italics (is correct), but C. testudinis must not be italicized (due to the rank as Cand.).
- The Discussion should be shortened due to the limited amount of data presented.
- The English is not correct in a number of sentences. The text should be checked by a native speaker or an experienced scientific writer.
Author Response
Thank you very much for the constructive comments to our manuscript. We have now implemented and improved the manuscript as follows:
- As a major objection, the term "species" is overused in the manuscript. (Prior to classification as a species, the term "taxon" may also be appropriate.) The authors use the term without having verified at least some of the criteria for species definition set out by the Subcommittee for the taxonomy of chlamydiae. For instance, sequence similarity values of the rDNA sequences to some of the related Chlamydia should be presented. There may also be observations on the tissue samples examined by the pathologist that could have revealed typical features of a chlamydial colonization.
Response to the reviewer: The manuscript has been amended, accordingly. The pathologist observation on the lung tissue was not possible because the specimen were autolytic.
The text was modified as follow:
"Histological data on lung tissues are not available because specimen were autolytic and their conservation status was not ideal for this analysis."
- Line 65: The description of the current status of the family Chlamydiaceaeis incomplete. The genus Chlamydia currently comprises 18 species, among them four at Candidatus In addition, the genus Chlamydiifrater (Cf.) with the two species Cf. phoenicopteri and Cf. volucris was recently introduced (see paper by Vorimore et al. Syst Appl Microbiol. 2021Jul;44(4):126200). Please, update the text.
Thank you for pointing this out, we updated the text, accordingly.
- Line 80: The word "seldom" should be deleted.
Amended accordingly.
- Line 81: Candidatus in Italics (is correct), but C. testudinis must not be italicized (due to the rank as Cand.).
Amended accordingly.
- The Discussion should be shortened due to the limited amount of data presented.
Response to the reviewer: Thank you for pointing this out. We are aware of the limited amount of data presented; however, we thought they are interesting to be shared within the Chlamydia research community. Unfortunately, we could not shorten the discussion according to your revision, because the Editor asked us to increase the length of the paper, according to the Special Issue guidelines (3000-4000 words).
- The English is not correct in a number of sentences. The text should be checked by a native speaker or an experienced scientific writer.
The manuscript has been revised by a native English speaker.
Reviewer 2 Report
Dear Authors,
your manuscript reports a very interesting study on Chlamydia infection in Trachemys scripta. A novel species was detected increasing the value of your investigation. I have only a curiosity: why did not you carry out histological examinations on the lung specimens?
Please, check the italic forms: somewhere they are not present.
Author Response
Thank you very much for the constructive comments to our manuscript. We have now implemented and improved the manuscript as follows:
I have only a curiosity: why did not you carry out histological examinations on the lung specimens?
Response to the reviewer:
Thank you for your revision work and for the comment. To satisfy your curiosity, authors have modified the text in the results to explain that the histological examination could not be performed on the lung tissues because the specimen were autolytic and stored incorrectly to achieve a valuable histological examination.
Please, check the italic forms: somewhere they are not present.
Authors amended the italic forms whenever needed.
Reviewer 3 Report
After reading the whole article, I have a few comments to the article that should be taken into account before its possible publication:
line 62: at the moment there is a mistake in the word "Enterobacteriaceae",
lines 86-95: this text should not appear in this place as it is introduction. It should be changed so that it does not constitute a summary, which should be at the end of the article - closer to the "Conclusions" section.
line 86: I am wondering about the number of samples - is it possible to infer the importance of Chlamydia for turtles on the basis of 4 pieces? It is customary to carry out research on many samples, or collect them over several years.
line 106: please indicate the author of the method to determine "Nucleic acids extraction"
line 181: the Authors indicated that "The amplification of the full ompA and 16S rRNA genes was possible for only one of the samples ". Please explain what was the reason for this.
lines 195-196: this sentence does not explain much, so I am asking you to elaborate on it or to refer to examples why the Authors of the text think so.
lines 225-226: the sentence is not clear and, in my opinion, undermines the purpose of the research. I am asking for its transformation, so that it is readable in interpretation.
There is no separate "Conclusions" section in the text, so I recommend creating it.
References:
1. There are two very old items (line 271 - from 1986 year and line 336 - from 1994 year). Please, if possible of course, replace them with newer ones.
2. On lines 317, 353 and 354-355 the titles of the articles are written in capital letters. Please check if this is correct and correct - if not.
3. Please check in the titles of the articles cited that the names of organisms (usually written in brackets) should not be in italics there.
Author Response
Thank you very much for the constructive comments to our manuscript. We have now implemented and improved the manuscript as follows.
- line 62: at the moment there is a mistake in the word "Enterobacteriaceae".
Modified accordingly.
- lines 86-95: this text should not appear in this place as it is introduction. It should be changed so that it does not constitute a summary, which should be at the end of the article - closer to the "Conclusions" section.
Response to the reviewer: The Authors agree with the comment and modify the text, accordingly. The lines were partially removed.
- line 86: I am wondering about the number of samples - is it possible to infer the importance of Chlamydia for turtles on the basis of 4 pieces? It is customary to carry out research on many samples, or collect them over several years.
Response to the reviewer: The Authors are aware of the limited amount of specimen examined. However, our findings aim at promoting new investigations and sampling of a larger amount of turtles in order to highlight the presence and importance of this pathogen in turtles in Italy.
- line 106: please indicate the author of the method to determine "Nucleic acids extraction"
Response to the reviewer: The procedure described in “Nucleic acid extraction” has been validated and used in our laboratory by means of reference materials (chlamydial strains) and authors followed the manufacturer’s instructions of the commercial kit (High Pure PCR Template Preparation Kit - Roche Diagnostics), as reported in the text.
- line 181: the Authors indicated that "The amplification of the full ompA and 16S rRNA genes was possible for only one of the samples ". Please explain what was the reason for this.
Response to the reviewer: Following the reviewer’s suggestions, the Authors have rephrased this section as follow in the hope it is now more straightforward:
"The complete amplification of the ompA and 16S rRNA genes was performed on both positive samples. However, full sequences were achieved only for the sample with a Cq value of 35.4, while for the other sample the Cq value was likely to be too low to obtain full sequences. "
- lines 195-196: this sentence does not explain much, so I am asking you to elaborate on it or to refer to examples why the Authors of the text think so.
Modifications have been made, according to the reviewer’s wish. The text was modified as follow:
"This new Chlamydia taxon is most closely related to C. pneumoniae, a strain with a known zoonotic potential, with a percentage of identity for the ompA gene sequence of 75.4%, and to other Chlamydia spp. found in reptiles and with unknown pathogenicity, C. serpentis (73.43%) and Candidatus C. sanzinia (71.9%) and C. corallus (73.7%). Moreover, there is high similarity (between 72% and 74%) with the clade composed by C. pecorum and sequences identified in Mitura et al. [27] that have been classified in the proposed Candidatus C. testudinis [28]."
- lines 225-226: the sentence is not clear and, in my opinion, undermines the purpose of the research. I am asking for its transformation, so that it is readable in interpretation.
Response to the reviewer: The Authors agree with the comment and have rephrased the statement for sake of clarity.
- There is no separate "Conclusions" section in the text, so I recommend creating it.
Response: The Authors have amended according to the reviewer’s suggestion.
- References:
1. There are two very old items (line 271 - from 1986 year and line 336 - from 1994 year). Please, if possible of course, replace them with newer ones.
Response: The Authors are aware that these items are very old; however, they cannot be replaced, since they include historical information on the Chlamydiaceae family.
- On lines 317, 353 and 354-355 the titles of the articles are written in capital letters. Please check if this is correct and correct - if not.
Amended accordingly.
- Please check in the titles of the articles cited that the names of organisms (usually written in brackets) should not be in italics there.
Modified, accordingly.